# Cybersecurity Threats and Their Mitigation Approaches Using Machine Learning—A Review

**Mostofa Ahsan [1,\*], Kendall E. Nygard [1], Rahul Gomes [2,\*], Md Minhaz Chowdhury [3], Nafiz Rifat [1] and Jayden F Connolly [2]**

1   Department of Computer Science, North Dakota State University, Fargo, ND 58102, USA; kendall.nygard@ndsu.edu (K.E.N.); nafiz.rifat@ndsu.edu (N.R.)
2   Department of Computer Science, University of Wisconsin-Eau Claire, Eau Claire, WI 54701, USA; flippcjt5179@uwec.edu
3   Department of Computer Science, East Stroudsburg University of Pennsylvania, East Stroudsburg, PA 18301, USA; mchowdhur1@esu.edu
\*   Correspondence: mostofa.ahsan@ndsu.edu (M.A.); gomesr@uwec.edu (R.G.)

**Abstract:** Machine learning is of rising importance in cybersecurity. The primary objective of applying machine learning in cybersecurity is to make the process of malware detection more actionable, scalable and effective than traditional approaches, which require human intervention. The cybersecurity domain involves machine learning challenges that require efficient methodical and theoretical handling. Several machine learning and statistical methods, such as deep learning, support vector machines and Bayesian classification, among others, have proven effective in mitigating cyber-attacks. The detection of hidden trends and insights from network data and building of a corresponding data-driven machine learning model to prevent these attacks is vital to design intelligent security systems. In this survey, the focus is on the machine learning techniques that have been implemented on cybersecurity data to make these systems secure. Existing cybersecurity threats and how machine learning techniques have been used to mitigate these threats have been discussed. The shortcomings of these state-of-the-art models and how attack patterns have evolved over the past decade have also been presented. Our goal is to assess how effective these machine learning techniques are against the ever-increasing threat of malware that plagues our online community.

**Keywords:** cybersecurity; machine learning; neural networks; classification; clustering; intrusion detection

## 1. Introduction

With the rapidly increasing prominence of information technology in recent decades, various types of security incidents, such as unauthorized access [1], denial of service (DoS) [2], malware attacka [3], zero-day attacks [4], data breaches [5], social engineering or phishing [6], etc., have increased at an exponential rate in the last decade. In 2010, the security community documented less than 50 million distinct malware executables. In the year 2012, this reported number doubled to around 100 million. From the record according to AV-TEST statistics, the security industry detected over 900 million malicious executables in 2019, and this number is rising [7]. Cybercrime and network attacks can result in significant financial losses for businesses and people. For example, according to estimates, an average data breach costs USD 3.9 million in the United States and USD 8.19 million globally [8], and cybercrime costs the world economy USD 400 billion per year [9]. The security community estimates [10], over the next five years, that the number of records broken will nearly quadruple. As a result, to minimize further losses, businesses must create and implement a comprehensive cybersecurity strategy. The most recent socioeconomic studies show that [11] the nation's security is dependent on governments, people with access to data, applications and tools that require high security clearance.

It is also dependent on businesses that give access to their employees, who possess the capacity and knowledge to identify such cyber-threats quickly and effectively. As a result, the primary concern that must be addressed immediately is to intelligently identify various cyber occurrences, whether previously known or unseen, and safeguard critical systems from such cyber-attacks adequately.

Cybersecurity refers to technologies and techniques that protect programs, networks, computers and data from being damaged, attacked or accessed by unauthorized people [12]. Cybersecurity covers various situations, from corporate to mobile computing, and can be divided into several areas. These are: (i) network security, which focuses on preventing cyber-attackers or intruders from gaining access to a computer network; (ii) application security, which considers keeping devices and software free of risks or cyber-threats; (iii) information security, which primarily considers the security and privacy of relevant data; and (iv) operational security refers to the procedures for handling and safeguarding data assets. Traditional cybersecurity solutions include a firewall, antivirus software or an intrusion detection system in network and computer security systems. Data science is driving the transformation, where machine learning, an essential aspect of "Artificial Intelligence", can play a vital role in discovering hidden patterns from data. Data science is pioneering a new scientific paradigm, and machine learning has substantially impacted the cybersecurity landscape [13,14]. As discussed in the article [15], with the advancement of technologies pertinent to launching cyber threats, attackers are becoming more efficient, giving rise to an increasing number of connected technologies. The graph in Figure 1 depicts timestamp data in terms of a specific date, with the *x*-axis representing the matching popularity and the *y*-axis representing the corresponding popularity in the range of 0 (minimum) to 100 (maximum). It is observed that the popularity values of cybersecurity and machine learning areas were less than 30 in 2015, and they exceeded 70 in 2022, i.e., more than double in terms of increased popularity. In this study, we focus on machine learning in cybersecurity, which is closely related to these areas in terms of security, intelligent decision making and the data processing techniques to deploy in real-world applications. Overall, this research is concerned with security data, using machine learning algorithms to estimate cyber-hazards and optimize cybersecurity processes. This project is also useful for academic and industrial researchers interested in studying and developing data-driven smart cybersecurity models using machine learning approaches.

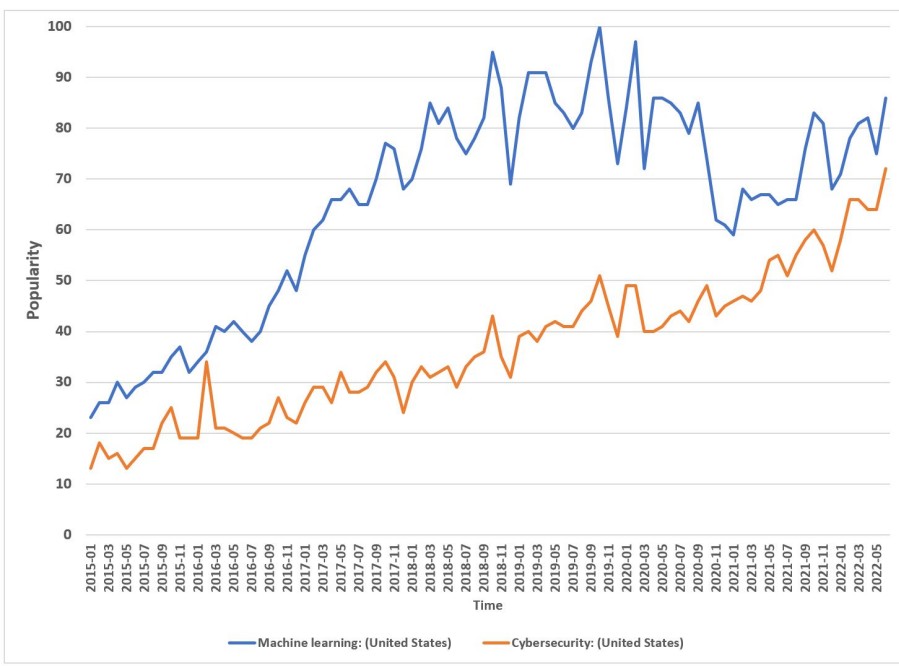

**Figure 1.** Google Trend for machine learning vs. data science vs. cybersecurity from 2015 to present.

Preventing cybersecurity attacks beyond a set of fundamental functional needs and knowledge about risks, threats or vulnerabilities requires analyzing cybersecurity data and building the right tools to process them successfully. Several machine learning techniques, which include but are not limited to feature reduction, regression analysis, unsupervised learning, finding associations or neural network-focused deep learning techniques, can be used to effectively extract the insights or patterns of security incidents. This is briefly discussed in the "Machine learning techniques in cybersecurity" section. These learning techniques can detect anomalies or malicious conduct and data-driven patterns of related security issues and make intelligent judgments to avert cyber-assaults.

Machine learning is a partial but significant departure from traditional well-known security solutions, including user authentication and access control, firewalls and cryptography systems, which may or may not be effective in meeting today's cyber business needs [16–18]. The critical difficulty is that domain experts and security analysts fix these manually in situations where ad hoc data management is required [19]. However, as a growing number of cybersecurity incidents in various formats are emerging over time, traditional solutions have proven ineffective in managing these cyber-hazards. As a result, a slew of new, complex attacks emerges and spreads rapidly over the network. Thus, several academics apply diverse data analytic and knowledge extraction models to create cybersecurity models, which are covered in the section "Machine learning techniques in cybersecurity", based on the efficient identification of security insights and the most recent security trends that may be more relevant. According to research, addressing the cyber problem necessitates the development of more flexible and efficient security systems that can adapt to attacks and update security policies to eradicate them on a timely basis intelligently. To do this, a huge amount of relevant cybersecurity data collected from different sources, such as network and system sources, must be analyzed. Moreover, these techniques should be implemented in a way that increases automation, with minimal to no human intervention.

The discussions of this study are listed below.

- To comprehend the applicability of data-driven decision making, a report on the existing idea of cybersecurity protection plans and associated approaches is presented first. To do this, several machine learning techniques used in cybersecurity have been discussed and numerous cybersecurity datasets, emphasizing their importance and applicability in this domain, have been presented.
- In addition, an examination of several related research challenges and future objectives in the field of cybersecurity machine learning approaches has been presented.
- Finally, the most common issues in applying machine learning algorithms on cybersecurity datasets have been explored within the scope of improvements to build a robust system.

The rest of this research is structured as follows. Section 2 describes the motivation for our research and provides an overview of cybersecurity technologies. Next, it defines and briefly covers numerous types of cybersecurity data, including cyber incident data. A brief discussion of different categories of machine learning techniques and their relationships with various cybersecurity tasks is presented in Section 3. The section also summarizes a number of the most effective machine learning algorithms for cybersecurity models in this domain. Section 4 briefly covers and emphasizes different research concerns and future directions in cybersecurity. Finally, an emphasis on some crucial elements of our findings has been presented in Section 5.

## 2. Background

Information and Communication Technology (ICT) infrastructure has significantly evolved over the last decade, and it is now widespread and thoroughly integrated into our modern civilization. As a result, today's security policymakers are urging ICT systems and applications to be protected from cyber-attacks [20]. Protecting an ICT infrastructure from various cyber-threats or attacks is referred to as cybersecurity [9]. Different aspects of

cybersecurity are associated with it, such as measures to protect ICT, the raw data and information it contains, as well as their processing and transmission. Other factors include the association of virtual and physical elements of the systems; the level of protection provided by these measures; and, finally, the associated field of professional endeavor [21]. According to [22], cybersecurity consists of different tools, guidelines and practices and is employed to protect software programs, computer networks and data from attack, unauthorized access or damage [22]. Research in [12] indicated that cybersecurity uses different processes and technologies that are useful to protect networks, programs, computers and data against assaults, unlawful access and destruction. In a nutshell, cybersecurity is concerned with the identification of various cyber-attacks and the implementation of appropriate defense tactics to protect the properties indicated below [22–24].

- Confidentiality is a property that prevents information from being shared with unauthorized entities, people or systems.
- Integrity is a property that protects data from being tampered with or destroyed without permission.
- Availability is used to ensure that authorized entities have timely and reliable access to information assets and systems.

### 2.1. Cyber-Attacks and Security Risks

There are three major security factors that are typically considered as risks: (1) attacks—who is attacking, vulnerabilities in the system; (2) the flaws or security pockets that they are attacking, and the impacts; (3) the consequences of the attack. These are all elements to consider [9]. A security breach occurs when information assets and systems' confidentiality, integrity or availability are endangered. Different forms of cybersecurity incidents might put an organization's or an individual's systems and networks at threat [9]; they can be grouped as follows.

Malware is malicious software that is designed to cause damage to a personal system, client, server or computer network [24]. Malware breaches a network by creating a vulnerable situation, such as a user clicking a dangerous link or email attachment and, hence, installing a risky software program. In most cases, the presence of such malicious software is not acknowledged by the authorized user(s) of the system. A system can become infected by malware in various ways. Examples include, but are not limited to, a victim being tricked into installing malware by opening a fake version of a legitimate file; a victim tricked in downloading malware by visiting malware-propagating websites; or a victim connecting to a malware-infected machine or device.

Malware victims can be any device containing computational logic. The victims can be end users and/or servers, their connecting network devices, process control systems, e.g., Supervisory Control and Data Acquisition systems. As with its victim types, malware can be of several types: bot executable, Trojan horses, spyware, viruses, ransomware and worms. Both the number and technologies of malware are growing fast. The most cost-effective solution is protecting the perimeter of the system by installing appropriate controls. Examples are intrusion detection/prevention systems (firewall, anti-virus software). With perimeter defense, an access control mechanism can control the access to a particular internal resource of the system. Despite these measures, there can be people violating their access rights. In such a situation, an organization policy on accountability can be implemented to punish a misdemeanor. Unfortunately, this combined effort of perimeter defense techniques with access control mechanisms and accountability may fail. Table 1 lists the recent defenses against malware attacks [24]. Typically, malware affects the network as follows:

- It blocks network key components.
- It installs additional harmful software for spying with malware itself.
- It gains access to personal data and transmits information.
- It disrupts certain components and makes the system inoperable for users.

**Table 1.** Defenses to protect data against malware.

| Defense Technology | Categories of Defense Technologies Used Against Malware | Description of Defense Categories |
|---|---|---|
| Cryptography is a way to change data in such a way that only the intended receiver has the information to extract information from the changed form (encrypted data). It is the most used method to secure data. | Identity-based cryptography [25] | This is a public key generated using identification-based information, e.g., email address. The generation is processed by a trusted certifying authority. This is an active research area, to overcome the inconveniences of this cryptography against malware attacks. |
| | Quantum cryptography [26] | In this cryptography, for the two parties, sender and receiver, the transmission generates cryptographic keys to encrypt data, following the laws of quantum mechanics. Hence, this encryption is not hackable. |
| Perimeter defense/defense in depth is securing an organization's network from outside intrusion | Firewall is the prevalent perimeter defense technology that controls network traffic (input data and outgoing data). It decides whether the data will go through or not based on a set of preset rules [27]. Despite the sophistication of firewalls, they can fail when a compromised but previously trusted system sends any request, and the attacking machine uses a trusted system's identity. | 1. Network-layer firewall or packet filtering works at the network layer controlling data flow but has the drawback of having static rules that are not able to block undesirable data. Hence, it cannot block malware payload. 2. Application-layer firewall controls the flow of input, output and system calls by an application. This firewall makes the tempering of internal components by malware difficult. 3. Proxy servers work as a mediator between outside connections and internal components of a system and hence can hinder the tampering of these components by malware. |
| | Network forensics [28] is the process of eavesdropping on the internet, Ethernet or TCP/IP to learn the attack pattern. There are numerous network forensics tools. | 1. eMailTrackerPro investigates the header of an email to look for an IP address, to find the sender. 2. Web browser traffic forensic tool, SmartWhoIs, can provide all available information about an IP address. 3. WebHistorian analyzes a website's URL. 4. Index.datanalyzer analyzes the browsing history, cache and cookies. 5. In the wireless LAN interface and network interface, packet intercepts can be caught using AirPcap and WinPcap, respectively. 6. Honeypots are mock resources that trap the attacker and gather information. |
| | Access control [29] differentiates between users and controls resource access of the user based on the user's preset rights. It provides authentication, authorization and accountability. | 1. Two broad divisions of access control, used in malware defense, are capability-based access control and the access control list-based approach. 2. Three access control models are Discretionary Access Control (DAC), Mandatory Access Control (MAC), Role-Based Access Control (RBAC). |

Ransomware blocks access to the victim's data and threatens the client with its destruction unless a ransom is paid. The Trojan horse is the most dangerous malware, which appears as useful and routine software and is mostly designed to steal financial information. A drive-day attack is a common method for distributing malware. These data require any action of a user to be activated. The users simply visit a benign-looking website and their personal system is infected silently and becomes an IFrame that redirects the victim's browser into a site controlled by the attacker.

Phishing is the practice of sending fraudulent communications or social engineering, which is mostly spread through emails. The goal is to steal the victim's data, such as credit card numbers and login credentials. As part of a larger operation as an advanced and persistent threat, this assault is frequently used to achieve a foothold in government or business networks. Spear phishing is targeted to particular individuals or organizations, governments or military intelligence to acquire trade secrets, financial gains or information. Whale phishing is mostly aimed at high-profile employees such as a CFO or CEO to gain vital access to a company's sensitive data.

Man-in-the-middle (MITM), also known as eavesdropping, occurs when the intruders successfully include themselves inside a two-party transaction or communication. The most common entries for MITM attackers are:

- Unsecured public WiFi, where intruders insert themselves between a visitor's device and the network.
- If an attacker's malware successfully breaches the victim's system, they can install software to gain the victim's secure information.

Denial-of-service (DDoS) is a type of attack that involves shutting down a network or service with a high volume of traffic to deplete resources and bandwidth, resulting in the system being unable to fulfill legitimate requests. DDoS attacks are frequently designed to target high-profile businesses' web servers, such as trading platforms, media, finance and government.

SQL injection (SQLI) aims to employ malicious code to manipulate back-end database access information that was not intended for display. Intruders could carry out an SQL injection simply by submitting malicious code into a vulnerable website search box.

A zero-day exploit attack refers to the threat of an unknown security vulnerability for which a fix has not yet been provided or about which the program developers are uninformed. To detect this threat, the developers require constant awareness.

DNS tunneling uses the DNS protocol to communicate non-DNS traffic over port 53 by sending HTTP and other protocol traffic over DNS. Since using DNS tunneling is a common and legitimate process, its use for malicious reasons is very often overlooked. Attackers can disguise outbound traffic as DNS, concealing data that are shared through an internet connection.

## 2.2. Defense Strategies

Defense strategies are required to protect data or information, information systems and networks from cyber-attacks or intrusions. They are in charge of preventing data breaches and security incidents, as well as monitoring and responding to threats, defined as any illegal activity that damages a network or personal system [30]. In this section, a popular perimeter defense strategy, the intrusion detection system, is presented. A detailed discussion on defense strategies can be observed in Figure 2.

An intrusion detection system (IDS) is described as "a software, device or application that monitors a systems or computer network for malicious activity or policy violations" [31]. User authentication, access control, anti-virus, firewalls, cryptography systems and data encryption are all well-known security solutions that may not be successful in today's cyber sector [16–18]. An IDS analyzes security data from numerous essential locations in a network or system to remedy the issues [32,33]. Furthermore, an IDS may detect both internal and external threats. Intrusion detection systems are classified into several groups based on their intended use. There are two major domains of IDS. One focuses on the intrusion detection techniques, and another focuses on the deployment or data source to which the IDS will be applicable. The deployment opportunities can be grouped into multiple research areas [34]. Two of the possible classifications could be the host-based intrusion detection system (HIDS), which monitors and analyzes data, files and secure information on a single system, and also the network intrusion detection system (NIDS), which monitors and analyzes network connections for suspicious activity. These two IDSs are able to scale based on the file system and network size. On the other hand,

misuse detection or signature-based IDS and anomaly-based IDS are most well-known intrusion detection systems used in theory [30]. Misuse detection is very effective against known attack types, which implies that it requires specific domain knowledge of intrusive incidents [35]. One of the most popular examples of misuse detection is SNORT.

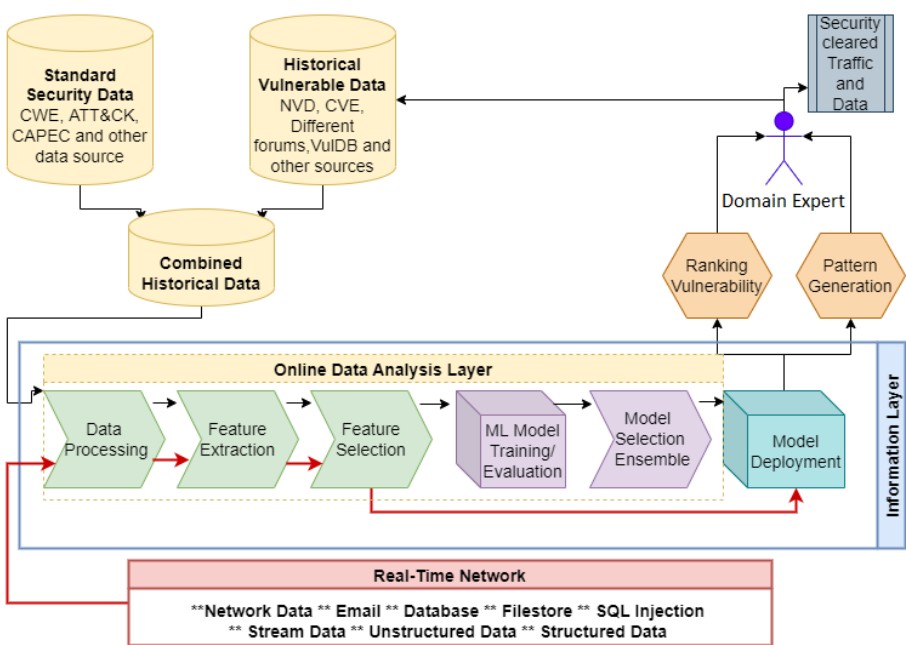

**Figure 2.** Flow chart of defense strategies in cybersecurity.

Signature-based detection works by signatures or fingerprint detection of network traffic [24]. This detection does not work properly for sophisticated and advanced malware that continuously evolves its patterns. This signature can be a pre-defined string, pattern or rule that correlates to a attack that has already occurred. A known pattern is defined as the detection of corresponding similar threats according to a signature-based intrusion detection system. An example of a signature-based IDS can be sequences used by mostly different types of malware, or known patterns or a byte sequence in a network traffic. Anti-virus software is used to detect these attacks, by identifying the patterns or sequences as a signature while performing a similar operation. As a result, a signature-based IDS is sometimes referred to as a knowledge-based or misuse detection system [36]. This technique can quickly process a large amount of network traffic, but it is firmly limited to rule-based or supervised detection. As a result, a signature-based system's most challenging difficulty is detecting new or unknown attacks using past knowledge.

Anomaly-based detection works by learning the pattern of normal network traffic and then flags the network traffic as anomalous if it is outside of this pattern [24]. The concept of anomaly-based detection is proposed to address the problems with signature-based IDSs that have been mentioned previously. The user activity and network traffic are first investigated in an anomaly-based intrusion detection system to discover dynamic trends, automatically create a data-driven model, profile normal behavior and detect anomalies during any departure [36]. As a result, an anomaly-based IDS is a dynamic approach that employs both supervised and unsupervised detection techniques. The capacity to detect zero-day assaults and wholly unknown threats is a significant advantage of anomaly-based IDS [37]. However, the identified anomaly or suspicious behavior sometimes leads to false alarms. Occasionally, it may identify several factors, such as policy changes or offering a new service, as an intrusion.

A hybrid detection approach [38,39] considers the anomaly-based and the signature-based techniques discussed above and can be used to identify intrusions. In a hybrid system, the signature-based detection system is used to identify known types of intrusions and an anomaly detection system is used for unknown attacks [40]. In addition to these

methods, stateful protocol analysis, which is similar to the anomaly-based method but employs established standard profiles based on agreed definitions of benign activity, can be used to detect intrusions that identify protocol state deviations [36]. A self-aware automatic response system would be the most effective of these options since it eliminates the requirement for a human link between the detection and reaction systems. There is a recent concept called Advanced Anomaly-Based Detection, which works by observing the network traffic for a certain duration [24]. Reinforcement learning (RL) is one of the advancements of Artificial Intelligence that can extend the logical reasoning of intrusion scenarios and prevent inexperienced attacks. The lack of cybersecurity attack data makes this technique extremely valuable to prevent the system against future attacks. Based on the agent or attack type, RL can be grouped into model-based and model-free approaches [41]. The application of machine learning techniques is extended in each of the branches of IDS. In the early period, it was only applicable for anomalous network data [42]. Later on, machine learning techniques were proven to be highly effective for the deployment of other IDS techniques on both the host and network domain [42]. With this observation, an adaptive and evolving model was built to deal with evolving malware signatures. Figure 3 summarizes the types of IDS based on detection and deployment.

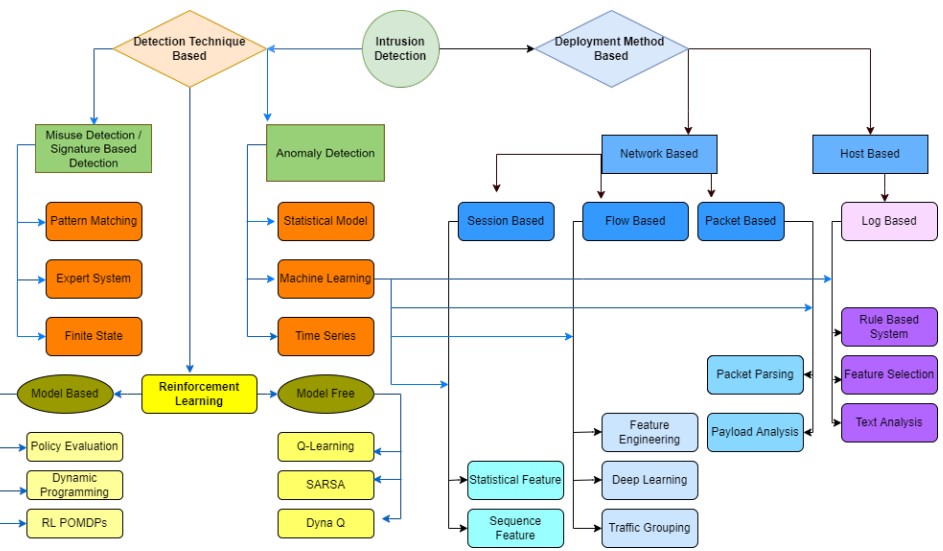

**Figure 3.** Types of intrusion detection systems.

### 2.3. Cybersecurity Framework

Cybersecurity forms an integral component of effective risk management in an organization. In order to handle cybersecurity risks, the role of NIST was modified to support the development of cybersecurity risk frameworks through the Cybersecurity Enhancement Act of 2014 [43]. This framework comprises three components, namely the framework core, implementation tiers and profiles. The framework core contains the basic guidance and standards that are imperative for an organization to manage risks posed by cyber threats. Implementation tiers proposed by NIST revolve around deciding on the scale of the proposed approach to mitigate threats. In other words, it allows an organization to understand the security standards required to guarantee protection. Finally, these frameworks support the creation of profiles that relate the cybersecurity activities to their respective outcomes. Profiles allow an organization to adapt their current approach to better suit demands.

The NIST framework is primarily divided into five functions [44]. These are identify, protect, detect, respond and recover. Identify revolves around the organization's capability to understand and effectively manage risks posed to assets such as data and physical devices by cyber threats. Protect is responsible for ensuring security mechanisms for the safe transmission of vital data and resources. Detect ensures that the organization is ready to implement techniques that can effectively recognize cyber threats. Respond ensures

that the organization is able to implement techniques that offer them the capability to respond to a threat. Finally, recover refers to activities that allow the organization to safely recover from a cybersecurity-related incident. Machine learning finds application in all these functions, especially in protect and detect. Protection categories such as access control can be implemented using machine learning. For example, NISTIR 8360 [45] utilizes a straightforward classification algorithm for verifying access control. Detect is perhaps the most widely explored area of machine learning. Almost all fields, such as anomaly detection and continuous monitoring, can benefit from a machine learning-based approached trained using a large amount of data. Discussion on machine learning techniques is given in the next section.

### 2.4. Cybersecurity Data

The availability of cybersecurity data drives machine learning in cybersecurity [46]. Machine learning techniques in cybersecurity are built on datasets, which are collections of records that contain information in the form of numerous qualities or features and related facts. As a result, it is vital to understand the nature of cybersecurity data, which encompasses a wide range of cyber events and critical components. The reasoning is that raw security data acquired from similar cyber sources can be used to investigate distinct patterns of security incidents. They can also be used to detect malicious behavior in order to develop a data-driven security model. This model will assist researchers to accomplish their objectives. In the field of cybersecurity, many datasets are available for various purposes, such as network intrusion analysis, malware analysis, phishing detection, fraud, anomalies or spam analysis. Numerous types of datasets have been outlined, including their varied aspects and incidents that are accessible on the internet, in Table 2, and we emphasize their use in diverse cyber applications based on machine learning techniques, analyzing and processing these networks effectively.

**Table 2.** Summary of cybersecurity databases.

| Dataset | Description |
|---|---|
| IMPACT [47] | Mostly known as the Protected Repository for the Defense of Infrastructures Against Cyber Threats (PREDICT), a community that produces security-relevant network operation data and research. Repository provides regularly updated network operations data of cyber defense technology development. |
| SNAP [48] | Not specific to security, but there are several relevant graph datasets. |
| KYOTO [49] | Traffic data from Kyoto University's Honeypots. |
| KDD'99 Cup [50] | Contains 41 features that could be used to evaluate ML models. Threats are categorized into four major target labels, such as remote-to-local (R2L), denial of service (DoS), probing and user-to-remote (U2R). |
| NSL-KDD [51] | Updated variant of KDD'99 Cup dataset. Records that are redundant have been removed. It also addresses issues associated with class imbalance. |
| DARPA [52] | LLDOS 1.0 and LLDOS 2.0.2 attack scenario data from the Authenticated Intrusion Detection System (IDS). MIT Lincoln Laboratory collects data traffic and threats from the DARPA dataset in order to evaluate network intrusion detection systems (NIDS). |
| UNSW-NB15 [53] | It has 49 independent features spread over nine different threat types, including DoS, which were gathered from the University of New South Wales (UNSW) cybersecurity Lab in 2015. UNSW-NB15 can be used for evaluation of ML-based anomaly detection systems in cyber applications. |
| ADFA IDS [54] | This is an intrusion dataset with different versions, named ADFA-LD and ADFA-WD, that is issued by the Australian Defense Academy (ADFA). This dataset is designed to evaluate host-based IDS. |

**Table 2.** *Cont.*

| Dataset | Description |
| --- | --- |
| MAWI [55] | A cybersecurity dataset regulated by Japanese network research institutions and academic institutions that is commonly used to detect and assess DDoS threats using machine learning techniques. |
| CERT [56] | The purpose of creating user activity logs was to validate insider-threat detection algorithms in this dataset. Based on machine learning, it can be used to track and evaluate user behavior. |
| Bot-IoT [57] | This is a dataset that includes authentic and simulated Internet of Things (IoT) network traffic, as well as various assaults for network forensic analytics in the IoT space. Bot-IoT is primarily used in forensics to assess reliability using multiple statistics and machine learning techniques. |
| DGA [58] | The Alexa Top Sites dataset reliably hosts domain names that are benign. Malicious domain names are collected from OSINT and DGArchive. These datasets find perfect application in DGA botnet detection or domain classification using automated ML models. |
| CTU-13 [59] | This is a labeled malware dataset including background traffic, botnet and normal user activities, which was captured at CTU University, Czech Republic. CTU-13 is used for data-driven malware analysis using machine learning techniques and to evaluate the standard malware detection system. |
| CAIDA [60] | The CAIDA'07 and CAIDA'08 datasets contain DDoS attack traffic and normal standard traffic history. They are primarily used to assess machine learning-based DDoS attack detection models and to spot internet DOS activities. |
| CIC-DDoS2019 [61] | The Canadian Institute for Cybersecurity has compiled a database of historical DDoS assaults. CIC-DDoS is an excellent network traffic behavioral analytics tool for detecting DDoS attacks using machine learning approaches. |
| ISCX'12 [62] | This dataset contains 19 features and 19.11% of the network traffic belongs to DDoS attacks. ISCX'12 was documented at the Canadian Institute for Cybersecurity and is well known for its use in the evaluation of the effectiveness of machine learning-based network intrusion detection modeling. |
| Malware [63] | This is a collection of malicious files from several malware-based datasets such as the Genome Project, VirusTotal, Virus Share, Comodo, Contagio, Microsoft and DREBIN. These datasets are commonly used for data-driven malware analysis and evaluation of existing malware detection systems utilizing machine learning techniques. |
| EnronSpam [64] | Email-based datasets are difficult to collect because of privacy concerns. This dataset is a collection of emails with spam and ham classification. |
| DREBIN [65] | Researchers have created these datasets from the Drebin project, which is publicly available, in order to encourage and improve research on Android malware. There are 5560 programs in this collection, spanning 179 different malware categories. The samples were collected between August 2010 and October 2012, and the MobileSandbox initiative made them freely available to cybersecurity practitioners. |
| CDX 2009 Network USMA [66] | This dataset highlights the correlation found between IP addresses associated with the PCAP files to hosts that are found on the internal USMA network. Not all network modifications are reflected in this dataset. |

## 3. Machine Learning Techniques in Cybersecurity

Machine learning (ML) is typically described as a branch of "Artificial Intelligence" that is closely related to data mining, computational statistics and analytics and data science, particularly focusing on allowing systems to learn from historical data [67,68]. As a result, machine learning models are often made up of a set of rules, procedures or complex functions and equations. These features can be used to uncover intriguing data patterns, recognize sequences or anticipate behavior [69]. As a result, ML could be useful in the field of cybersecurity. Figure 4 depicts a summarized view of the most frequently used machine learning techniques for cybersecurity. The taxonomy is primarily divided into three sections, namely deep learning models, shallow models and reinforcement learning.

Machine learning algorithms—in this case, shallow models—are further classified into and supervised learning and unsupervised learning. In supervised learning, the models usually do not have a dependent variable and mostly rely on the internal patterns available in the dataset to group the data into different categories. This can be achieved using different algorithms, such as K-means, Sequential Pattern Mining, DB scan [70] and the a

priori algorithm [71]. In supervised learning, the models usually have class labels to verify the predictions. Naïve Bayes, for example, uses probabilistic distribution to identify to which category a class label belongs. Decision trees create a tree-like structure based on a training set. For prediction, once the tree is built, any unknown record can be sorted based on the tree structure. Random forest [72] uses a similar approach, but instead of building one decision tree, it builds multiple decision trees and then uses a voting scheme to classify a record. Because of the collective nature of the decision-making process, random forest usually has higher classification accuracy. Support vector machine (SVM) [73] works by creating a linear decision boundary from the dataset. This can be compared to a binary classification. SVMs are also capable of transforming the data using a kernel trick. This allows SVMs to classify nonlinear datasets as well.

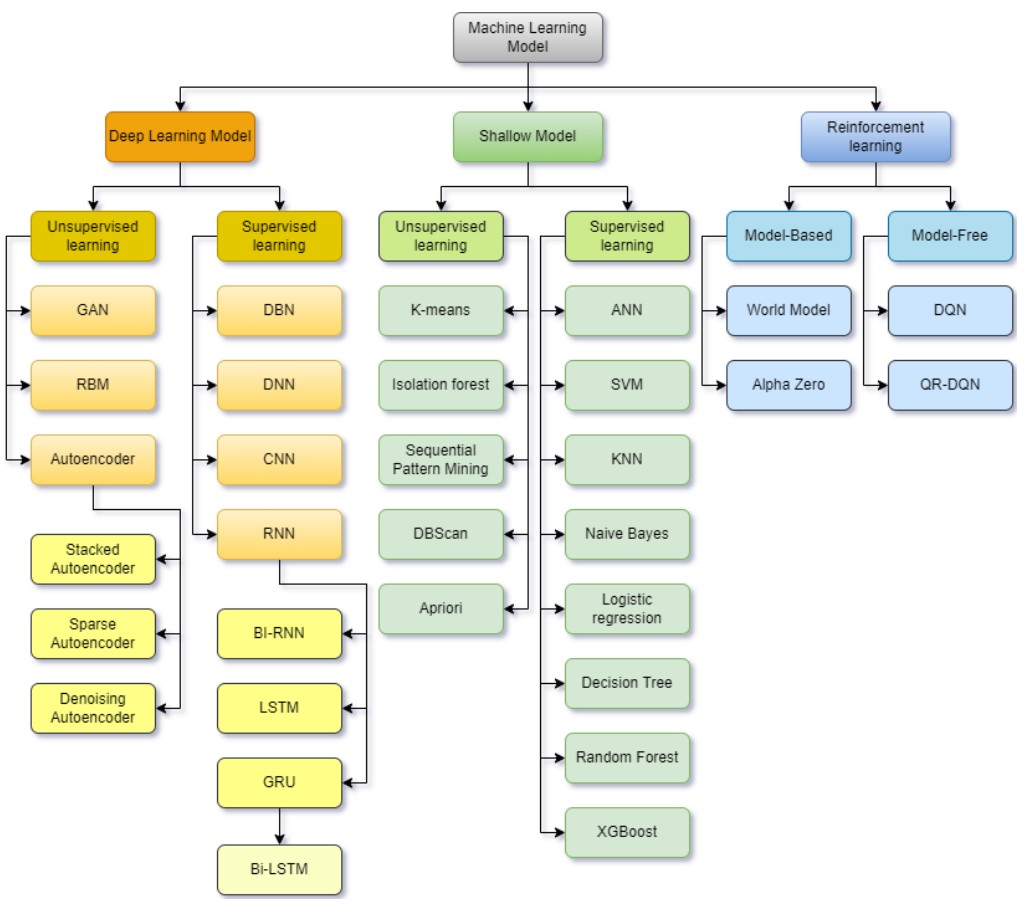

**Figure 4.** Taxonomy of machine learning algorithms.

Deep learning models can also classify or cluster algorithms. However, their approach is quite different from machine learning models. Unlike their counterparts, they do not have a fixed algorithm for prediction. Hence, they are also known as black box models because they analyze the data, identify patterns and use the patterns for production. Deep learning models use artificial neural networks, which are built using several perceptrons. These perceptrons are connected in a randomized fashion during the onset of model training. By looking at the data and training over a given time, the perceptrons gain values, also known as weights, that are better suited for classifying the dataset at hand. There are different varieties of deep learning models. Convolutional neural networks find application in classifying image data. They have also been used to classify cybersecurity datasets by transforming the data into a format that resembles an image. Recurrent neural networks (RNN) find application in classifying data that have a temporal aspect. Several improved versions of RNN include LSTM or long short-term memory, as well as Bi-LSTM. Unsupervised learning with deep learning includes autoencoders as well as generative

adversarial networks. Autoencoders mostly use feature reduction, where information is transformed into a compressed form before further processing. They are well adapted to compressing information in a meaningful way, thereby increasing the prediction accuracy.

Reinforcement learning explores a different approach of training a model to differentiate between long-term and short-term goals. Agents interact with their environment and, based on their actions, they are rewarded or penalized. The reward is variable, thereby teaching the model to become better. A popular example is DQN [74] or Deep Q Networks. In DQN, the mapping function between states and actions is accomplished using deep learning, thereby reducing the requirement for a large table of Q-learning (TQL). A variant of DQN is QR-DQN [75], which uses quantile regression to model the possible distributions, instead of returning the mean distribution. This can be compared to the difference between decision trees and random forest, summarized above.

In this section, different methods that can be used to solve machine learning techniques and how they are related to cybersecurity are explored. The traditional machine learning models are often known as shallow models for intrusion detection systems (IDS). Some of these techniques have been researched very extensively, and their approach is well known. They concentrate on tasks other than intrusion detection, and include tasks such as labeling, efficiently detecting attacks, as well as the optimal management of available and processed data.

### 3.1. Stages of a Cyber-Attack

Organizations can assess the cybersecurity risk to them and can identify certain security threats. They can then implement security controls or measures against these threats. They can utilize the National Institute of Standards and Technology (NIST) Special Publications, although they may not be a US federal agency or related contractor [76]. NIST Special Publications provide step-by-step guidance for applying a risk management framework to federal information systems. In this guidance, a set of security issues are identified and common controls or measures against these security issues/threats are listed. In a recent study, machine learning tools were suggested as efficient controls or measures [77]. Such measures can be applicable to all five phases of a cyber-attack.

There are five phases of a cyber-attack. They are reconnaissance, scan, attack (denial-of-service attacks, gain access using application and operating system attacks, network attacks), maintain access (using Trojans, backdoors, rootkits, etc.) and cover tracks and hiding. An interruption at any phase can either interrupt or halt the entire process of attack. Machine learning algorithms can be used in all of these phases to help fight against cyber-attacks by disrupting the attacker's workflow.

During the reconnaissance or preparation phase of the attack, an adversary uses techniques such as a social engineering attack (phishing, malicious call, etc.). Machine learning algorithms can look for email signatures and detect malicious or phishing email signatures and block them. There are cases when an attacker calls the target organization and impersonates a third party to obtain valuable information (known as "voice phishing" or "vishing"). Call source analysis using machine learning algorithms can flag and block such calls. Another example use of machine learning is scanning any external devices connected to the organization's property, e.g., a USB device. Such a scan prevents malicious software from propagating through such devices. Another example is when the adversary wishes to guess the access password to obtain unauthorized access (violating confidentiality) [78]. Rule-based machine learning algorithms can detect the most common passwords that are used by the organization's employees and can recommend a list of unrecommended passwords. This will hinder the reconnaissance step. Such machine learning algorithms can be placed in strategic locations, e.g., key machines and networks.

During the scan phase, sometimes called "Weaponization", the cyber-attacker or adversary exploits the vulnerabilities of the target system. The attacker uses automated tools, such as Metasploit, AutoSploit and Fuzzers [78]. Machine learning algorithms can be used to automatically scan and find the vulnerabilities by an ethical hacker before the

adversary can. For example, a machine learning-based penetration test can be implemented, specifically by integrating the algorithms into the penetration testing tools, e.g., Metasploit. Such algorithms, upon being used by a pen tester, can find novel weaknesses.

Machine learning algorithms are a strong measure against the attacks (phase 3 of cyber-attack). Machine learning algorithms that can be used to provide cybersecurity are linear regression, polynomial regression, logistic regression, naïve Bayes classifier, support vector machine, decision tree, nearest neighbor, clustering, dimensionality reduction, linear discriminant analysis and boosting [4]. The applications of these algorithms as a measure against cybersecurity problems are spam detection (includes phishing), malware detection, denial-of-service attacks (including DDoS) and network anomaly detection. Other forms of attacks are associated with biometric recognition, authentication, identity theft detection and social media analytics. Information leakage detection, APT or advanced persistent threat detection, hidden channel detection and software vulnerability detection are also some modern threats that need addressing.

During phase four of a cyber-attack, malware is used to maintain access by the attacker, e.g., Trojans, backdoors or rootkits. Machine learning algorithms can detect such malware traffic packets when the malware contacts the attacker and vice versa. For example, for malware detection, support vector machines (SVM) are an efficient option [79]. SVM was implemented using Weka to detect Android OS malware (260 samples), using static features analysis. Here, a black box method was used by analyzing the malware's behavior rather than executing the malware. In the first step, a Python code was used to extract Android application packages' (APK) features, one package at a time. Both malicious (201 samples) and benign (59 samples) APKs were selected. In the second step, an SVM classifier (Weka and LibSVM classifier) was trained by these features, to identify malware from these APKs. In the testing phase, the used APKs were downloaded from the repositories: Android, Malware Dataset, Kaggle, Drebin, Android Malshare and APKPure. The receiver operating characteristic or ROC curve was used to present the result. An enhancement of this application used the dynamic features of malware, as malware keeps changing its features. An SVM model can be trained to perform binary classification from a set of features of network packets. The trained classifier can detect a DDoS attack by identifying normal vs. abnormal network traffic, especially for IoT devices. Examples of the features used to train machine learning algorithms include the destination IP address, sequence number and minimum, maximum and average packets for each destination IP address, received signal strength indication, network allocation vector, value injection rate and inter-arrival time between consecutive frames, etc. The traffic information was collected by placing sensors at significant points of the network, e.g., at the gateway level, for a traffic session of 15–20 min. This classifier can be used as an extra security layer in IDS.

Another example is the application of various clustering techniques (K-means, DB-SCAN and Hierarchical) [79]. Clustering is useful for malware detection, phishing attack detection, spam filtering and detecting the larger family of software bugs known as side-channel attack detection. In [79], both malware and goodware Android APKs were installed in an Android emulator. Then, their resource usage statistics (CPU and RAM) were recorded for all three clustering techniques. For all three clustering algorithms, a total of 217 data instances were used, with 145 for training and 72 for testing. The conclusion was that CPU-RAM usage statistics are not an efficient feature for clustering malware and goodware. The nearest neighborhood (NN) search is used in access control. For example, an NN is used to identify actual vs. forged biometrics (e.g., fingerprints) through classification based on their patterns [79]. The CSD200 model was used as the fingerprint scanner to take 100 samples (10 people, total 100 fingers). MATLAB was used to convert these images into an array or matrix. Such a machine learning algorithm can automatically make decisions as to whether a biometric is forged or original. In [79], decision trees were used, e.g., Iterative dichotomizer 3 (ID 3) and its successor, C4.5, to identify malware efficiently. The dataset used was from the Cardiff University Research Portal. In another research work [80], anomalous services running into the computer systems, both offline and online, were iden-

tified using a neural network-based model (NARX-RNN), AI-based multi-perspective SVM, principal component analysis (PCA) and hierarchical process tree-based reinforcement learning techniques.

During phase five or the covering tracks phase, the attacker wishes to confirm that their identification is not being tracked. They employ different techniques, including corrupting machine learning tools' training data to misidentify their data. The machine learning algorithms themselves can be robust but their training data may not be. Deceptive training data make the algorithm inefficient. This process of forging training data is called adversarial machine learning (AML). The severity is serious for cybersecurity applications. The countermeasures against such polluted data include game theory (non-cooperative game/Nash equilibrium, Zero-Sum Versus Non-Zero Sum Game, simultaneous move vs. sequential game or Bayesian Game) [81]. An example of AML is network traffic classification. Performing deep packet inspection is hard when the traffic payload is encrypted [82]. For such traffic, it is possible that the machine learning classifier (e.g., network scanning detector) is deceived by an adversary, to tag malware or botnet communications or NMap network scanning traffic as benign. It is possible that the adversary can mimic the features of benign traffic and can infer the classification output. What happens if the adversary's traffic is classified as malicious? The adversary does not obtain any feedback but their traffic will probably be blocked. This will give them an indication that their traffic has been classified as malicious and prompt the adversary to change the traffic signature. Improved machine learning techniques exist that can work as a measure against adversarial attacks [83]. For example, an activation clustering method was introduced that identifies the hidden layer of a deep neural network where an adversarial trigger lies. Using empirical learning algorithms, poisonous data points can be identified when poisoning attacks happen against an SVM.

### 3.2. Supervised Learning

Supervised learning relies on useful information in historical labeled data. Supervised learning is performed when targets are predefined to reach from a certain set of inputs, i.e., task-driven approach. Classification and regression methods are the most popular supervised learning techniques [84]. These methods are well known for classifying or predicting the target variable for a particular security threat. For example, classification techniques can be used in cybersecurity to indicate a denial-of-service (DoS) attack (yes, no) or identify distinct labels of network risks, such as scanning and spoofing. Naive Bayes [85], support vector machines (SVM) [86], decision tree [87,88], K-nearest neighbors [89], adaptive boosting [90] and logistic regression [91] are some of the most well-known classification techniques in shallow models.

Naive Bayes finds a good amount of use in cybersecurity. The authors in [92] used the naive Bayes classifier from the Weka package and KDD'99 data for training and testing. Data were grouped into the four attack types (probe and scan, DoS, U2R and R2L) and their classifier achieved 96%, 99%, 90% and 90% testing accuracy, respectively. The cumulative false positive rate was 3%. The authors in [93] developed a framework using a simple form of Bayesian network using the KDD'99 data and used categories to depict different attack scenarios. Solving an anomaly detection problem, the reported results were 97%, 96%, 9%, 12% and 88% accuracy for normal, DoS, R2L, U2R and probe or scan categories, respectively. The false positive rate was not reported but can be inferred to be less than 3%. Naive Bayes was also used as one of the methods in [94] to solve a DoS problem, which attempted to resolve the botnet traffic in filtered Internet Relay Chat (IRC), therefore determining the botnet's existence and origin. The study conducted used TCP-level data that were collected from 18 different locations on the Dartmouth University campus' wireless network. This data collection occurred over a span of four months. A filter layer was used to extract IRC data from the network data. Labeling was a challenge so the study utilized simulated data for the experiments. The performance of the Bayesian network showed 93% precision with a false positive rate of 1.39%. C4.5 decision trees were also used for comparison

and achieved 97% precision, but the false positive rates were higher, at 1.47% and 8.05%, respectively.

In [95], the authors used an SVM classifier to detect DDoS attacks in a software-defined network. Experiments were conducted on the DARPA dataset, comparing the SVM classifier with other standard classification techniques. Although the classifier had higher accuracy, the SVM took more time, which is an obvious flaw. In [96], the authors used a least-squares SVM to decrease the training time on large datasets. Using three different feature extraction algorithms, they reduced the number of features from 41 to 19. The data were resampled to have around 7000 instances for each of the five classes of the KDD'99 dataset. Overall, the classification was reported at 99% for DoS, probe or scan, R2L and normal classes and 93% for U2R. Research in [97] utilized a robust SVM, which is a variation of SVM where the discriminating hyperplane is averaged to be smoother and the regularization parameter is automatically determined. Preprocessing training and testing of data was done on the Basic Security Module from the DARPA 1998 dataset. It showed 75% accuracy with no false positives and 100% accuracy with 3% false positives.

In [98], the authors utilized decision trees to generate detection rules against denial-of-service and command injection attacks on robotic vehicles. Results showed that different attacks had different impacts on robotic behavior. A decision tree-based intrusion detection system that may change after intrusion by analyzing the behavior data through a decision tree was implemented in [99]. The model was used to prevent advanced persistent threat (APT) attacks, which use social engineering to launch various forms of intrusion attacks. The detection accuracy in their experiments was 84.7%, which is very high for this experiment. Decision trees were also used in [100], where the authors replaced the misuse detection engine of SNORT with decision trees. SNORT is a known tool that follows the signature-based approach. The authors utilized clustering of rules and then derived a decision tree using a version of an ID3 algorithm. This rule clustering reduced the number of comparisons to determine which rules were triggered by given input data, and the decision tree located the most discriminating features of the data, thereby achieving parallel feature evaluation. This method achieved superior performance to that of SNORT when applied to the 1999 DARPA intrusion detection dataset. The results varied drastically depending on traffic type. The fastest were up 105%, with an average of 40.4% and minimum of 5% above the normal detection speed of SNORT. The number of rules was also increased from 150 to 1581, which resulted in a pronounced speed-up versus SNORT.

Ensemble learning techniques such as random forest (RF) have also been explored in cybersecurity research. Random forest uses multiple decision trees to draw a conclusion and can be more accurate than a single decision tree. In [101], the authors employed a random forest algorithm on the KDD dataset for misuse, anomaly and hybrid-network-based intrusion detection. Patterns were created by the random forest and matched with the network for misuse detection. To detect anomalies, the random forest detected novel intrusions via outliers. Using the patterns built by the model, new outliers were also discovered. The study implemented a full system solution including anomaly detection. Data were grouped into majority attacks and minority attacks. This hybrid system achieved superior performance for misuse detection, where the error rate on the original dataset was 1.92%, and it was 0.05% for the balanced dataset.

Regression algorithms are useful for forecasting a continuous target variable or numeric values, such as total phishing attacks over a period of time or network packet properties. In addition to detecting the core causes of cybercrime, regression analysis can be utilized for various risk analysis forms [102]. Linear regression [67], support vector regression [86] and random forest regressor are some of the popular regression techniques. The main distinction between classification and regression is that, in regression, the output variable is numerical or continuous, but in classification, the projected output is categorical or discrete. Ensemble learning is an extension of supervised learning that mixes different shallow models, e.g., XGBoost and random forest learning [72], to complete a specific security task. A summary of supervised approaches is shown in Table 3.

**Table 3.** Shallow machine learning algorithms Used in cybersecurity.

| Algorithm | Objective | Dataset | Accuracy | Reference |
|---|---|---|---|---|
| Naive Bayes | Can be used to analyze continuous and discrete values. Features are evaluated in a mutually exclusive fashion, making it relatively fast, thereby finding applicability in real-time decision making. | KWeka package, KDD 1999 | 90–99% | [103] |
| | | KDD 1999 | 97% | [93] |
| | | TCP data collected from the Dartmouth University campus' wireless network | 93% | [94] |
| Support Vector Machines | Effective in high-dimensional spaces. Relatively memory-efficient. Numerical and categorical features. | DARPA | 95.11% | [95] |
| | | KDD-99 | 93–99% | [96] |
| | | DARPA 1998 | 75–100% | [97] |
| Decision Tree | Requires little data preparation. Can be used to analyze continuous and discrete data. Can be generalized using dynamic tree-cut parameters. | TCP data collected from the Dartmouth University campus' wireless network | 97% | [94] |
| | | 3000 behavior events collection | 84.7% | [99] |
| | | KDD dataset | 94.7% | [101] |
| Sequential Pattern Mining | Frequent sequential patterns for a frequency support measure. | DARPA 1999 and 2000 | 93% | [104] |
| DBSCAN | Identify outliers, separate clusters of high density from clusters of low density. | KDD-99 | 98% | [105] |
| ADMIT | Not reliant on a lot of labeled data. Uses a dynamic clustering technique Modified form of K-means clustering. | Data collected from UNIX users from Purdue University | 80% | [106] |
| A priori algorithm | The resulting rules are intuitive. Does not require labeled data as it is fully unsupervised. | Nine different-sized custom databases | 70–100% | [107] |
| Radial Basis Function (RBF) | Real-time network anomaly detection. | KYOTO | 95.6% | [108] |
| Random forest | Multi-class classification of network traffic threats. | KDD'99 Cup | 99.0% | [109] |
| Extra-tree classifier (ETC) | Multi-class classification of DoS, probe, R2L and U2R. | KDD'99 Cup | 99.51% | [110,111] |
| Radial Basis Function (RBF) | Comparative classification between lazy, eager learning and deep learning. | DARPA | 97.41% | [108,112] |
| Random forest | Comparative classification between lazy, eager learning and deep learning. | UNSWNB15 | 95.43% | [113,114] |
| Random forest | Android malware detection. | DREBIN | 94.33% | [115–117] |
| XGBoost | Classification of spam and ham from emails. | ENRON Spam | 98.67% | [118–120] |

### 3.3. Unsupervised Learning

The main goal of unsupervised learning, also known as data-driven learning, is to uncover patterns, structures or relevant information in unlabeled data [121]. Risks such as malware can be disguised in a variety of ways in cybersecurity, including changing their behavior dynamically to escape detection. Clustering techniques, which are a form of unsupervised learning, can aid in the discovery of hidden patterns and insights in datasets, as well as the detection of indicators of sophisticated attacks. In addition, clustering algorithms can effectively spot anomalies and policy violations, recognizing and eliminating noisy occurrences in data, for example. K-means [122] and K-medoids [123] are clustering

algorithms used for partitioning, while single linkage [124] and complete linkage [125] are some of the most widely utilized hierarchical clustering methods in various application sectors. Furthermore, well-known dimensionality reduction techniques such as linear discriminant analysis (LDA), principal component analysis (PCA) and Pearson correlation, as well as positive matrix factorization, can be used to handle such problems [67]. Another example is association rule mining, in which machine learning-based policy rules can learn to avert cyber incidents. The rules and logic of an expert system are normally manually documented and implemented by a knowledge engineer working with a domain expert [30,121,126]. In contrast, association rule learning identifies the rules or associations among a set of security aspects or variables in a dataset [127]. Different statistical analyses are performed to quantify the strength of associations [102]. In the domain of machine learning and data mining, various association rule mining methods have been presented, such as tree-based [128], logic-based [129], frequent pattern-based [130–132], etc. Moreover, a priori [130], a priori-TID and a priori-Hybrid [130], AIS [127], FP-Tree [128], RARM [133] and Eclat [134] have been used extensively for association rule learning. These algorithms are able to resolve such difficulties by creating a set of cybersecurity policy rules.

The authors in [104] applied sequential pattern mining on the DARPA 1999 and 2000 data to reduce the redundancy of alerts and minimize false positives. Using the a priori algorithm, they discovered attack patterns and created a pattern visualization for the users with a support threshold of 40%. They were able to detect 93% of the attacks in twenty seconds. The study reported a real-time scenario with 84% attack detection, with the main variation coming from the support threshold. In [105], DBSCAN was used as a clustering method to group normal versus anomalous network packets in the KDD'99 dataset. The dataset was preprocessed and features were selected using correlation analysis. In this study, the performance was shown at 98% for attack or no-attack detection. The authors in [106] took a different data mining approach to create an intrusion detection system called ADMIT. This system does not rely on labeled data; rather, it builds user profiles incrementally using a dynamic clustering technique. It uses sequences, which are a list of tokens collected from users' data, to build a profile. These sequences are classified as normal or anomalous based on using a modified form of K-means clustering. They set a value of 20 for the maximum sequence length, which resulted in 80% performance accuracy with a 15% false positive rate. A unique algorithm based on the signature a priori algorithm was used to find new signatures of attacks from existing attack signatures, proposed in [107]. Here, the authors compared their algorithm processing time to that of the a priori and found that their algorithm was much faster. A summary of unsupervised approaches is also shown in Table 3.

### 3.4. Artificial Neural Networks (ANN)

ANN is a segment of machine learning, in the area of Artificial Intelligence, that is a computationally complex model inspired by the biological neural networks in the human brain [67]. The basic idea behind ANN was first introduced in the 1940s, with ANNs becoming a popular idea and technology in the late 1970s and continuing into the 1990s. Although SVMs were prominent in the 1990s, overshadowing the ANNs, they have received popularity recently and are steadily increasing in use. ANNs consist of multiple neurons that work together in layers to extract information from the dataset. The primary difference is the performance of ANN versus shallow machine learning as the amount of security data grows. The number of studies of ANN-based intrusion detection systems has increased rapidly from 2015 to the present. In [135], the authors utilized ANNs to detect misuse. Using data generated by a RealSecure network monitor, ten thousand events were collected, of which 3000 were simulated attacks by programs. Preprocessing of the data was performed and ten percent of the data were selected randomly for testing, with the rest used for training the ANN. The error rates for training and testing were 0.058 and 0.070, respectively. Each packet was categorized as normal or attack. A combination of keyword selection and ANN was proposed in [136] to improve IDS. Keyword selection was

performed on Telnet sessions and statistics were derived from the number of times that the keywords occurred (from a predefined list). These statistics were given as input to the ANN and the output identified the probability of an attack. The authors in [137] compared fuzzy logic and artificial neural networks to develop comprehensive intrusion detection systems and tested them using the KDD'99 dataset. They presented a detailed discussion on the preprocessing, training and validation of the proposed approach. The "class" characteristic in this dataset, which is made up of around 5 million data instances with 42 properties, determines whether a particular instance is a typical connection instance or one of the four types of attacks that need to be recognized. Five different machine learning approaches were compared, among which the FC-ANN-based approach [138] and the hierarchical SOM-based approach [139] were the best detectors.

Deep learning models, which are a form of ANN, learn feature representations directly from the original data, such as photos and texts, without the need for extensive feature engineering. As a result, deep learning algorithms are more effective and need less data processing. For large datasets, deep learning methods have a significant advantage over classical machine learning models. Some of the widely used deep learning techniques in cybersecurity include deep belief networks (DBNs), convolutional neural networks (CNNs) and recurrent neural networks (RNNs) as supervised learning models. Several variants of autoencoders, restricted Boltzmann machines (RBMs) and generative adversarial networks (GANs) have been used with success for unsupervised learning. The authors in [140] used DBNs to detect malware with an accuracy of 96.1%. The DBNs used unsupervised learning to discover layers of features and then used a feed-forward neural network to optimize discrimination. DBNs can learn from unlabeled data, so, in the experiments, DBNs provided a better classification result than SVM, KNN and decision tree. DBNs were also used in [141] to create an IDS. The proposed four-layer model was used on the KDD'99 Cup dataset, where the authors reported accuracy, precision and false acceptance rate (FAR) of 93.49%, 92.33% and 0.76%, respectively. In [142], a DBN-based ad hoc network intrusion detection model was developed with an experiment on the Network Simulator (NS2) platform. The experiment showed that this method can be added to the ad hoc network intrusion detection technology. Accuracy and FAR were reported as 97.6% and 0.9%, respectively. A deep learning approach called DeepFlow was proposed in [143] to directly detect malware in Android applications. The architecture consisted of three components for feature extraction, feature coarse granularity and classification. Two modules were used to assess malware sources from the Google Play Store. Experiments showed that DeepFlow outperformed SVM, ML-based algorithms and multi-layer perceptron (MLP). A summary of neural network approaches is shown in Table 4.

**Table 4.** Deep machine learning algorithms used in cybersecurity.

| Algorithm | Objective | Dataset | Accuracy | Reference |
|---|---|---|---|---|
| ANN | Abilities to learn, classify and process information; faster self-organization. | RealSecure network monitor | 96.5% | [135] |
| DeepFlow | Custom-developed to distinguish malware. It uses the static taint analysis tool FlowDroid. Identifies sensitive data flows in Android apps. | Features extracted from 11,000 benign and malicious apps from Google Play Store | 95.05% | [143] |
| DBNs | Discovers layers of features and uses feed-forward neural network to optimize discrimination. | Custom dataset | 96% | [140] |
| | | KDD'99 Cup | 93.49% | [141] |
| | | Network feature sample | 97.60% | [142] |

**Table 4.** *Cont.*

| Algorithm | Objective | Dataset | Accuracy | Reference |
|---|---|---|---|---|
| Deep Belief Network (DBN) | Real-time network anomaly detection. | KYOTO | 98% | [144] |
| Gated Recurrent Unit (GRU) | Multi-class classification of network traffic threats. | KDD'99 Cup | 98.64% | [145,146] |
| CNN-LSTM | Multi-class classification of DoS, probe, R2L and U2R. | KDD'99 Cup | 99.70% | [147–150] |
| Deep Feed Forward (DFF) | Comparative classification between lazy, eager learning and deep learning. | DARPA | 99.63% | [151,152] |
| Temporal convolutional networks (TCN) | Comparative classification between lazy, eager learning and deep learning. | UNSWNB15 | 99.6% | [153–155] |
| CNN | Android malware detection. | DREBIN | 99.29% | [156–158] |
| Bi-LSTM | Classification of spam and ham from emails. | ENRON Spam | 98.84% | [103,159] |

## 4. Future Improvements and Challenges for ML-Based Cybersecurity

Various research concerns and obstacles in machine learning in cybersecurity must be addressed while extracting insights from relevant data to make data-driven intelligent cybersecurity decisions. The issues in the following section range from data collection to decision making.

### 4.1. Cybersecurity Dataset Availability

In cybersecurity, source datasets are critical, as they are in machine learning. The majority of publicly available datasets are outdated and may not be sufficient in identifying the undocumented behavioral patterns of various cyber-attacks. Even though current data can be translated into knowledge after a series of processing steps, there is still a lack of understanding of the nature of recent attacks and their recurrence patterns. As a result, additional processing or machine learning approaches may result in a low accuracy rate when it comes to making final decisions. One of the fundamental obstacles in using machine learning techniques in cybersecurity is establishing a large number of recent cybersecurity datasets for particular issues such as attack prediction or intrusion detection.

### 4.2. Cybersecurity Dataset Standard

The cybersecurity datasets could be unbalanced, noisy, incomplete, irrelevant or contain inconsistent examples of a particular security violation. The quality of the learning process and the performance of machine learning-based models may be harmed by such issues in a dataset [160,161]. To build a data-driven solution for cybersecurity, such problems in data need to be addressed before the application of machine learning techniques. It is imperative to understand the problems in cybersecurity data and effectively address these issues using existing or novel algorithms to perform tasks such as malware and intrusion detection, among others. Some methods to solve these issues are associated with feature engineering [154], where model features are analyzed to remove correlated features. This technique reduces data dimensionality, thereby reducing complexity. Handling data imbalance is imperative, which can be done by utilizing hybrid models, as reported in [137], or generating synthetic data [162,163]. Other issues pertaining to data leakage should also be addressed.

### 4.3. Hybrid Learning

Signature-based intrusion detection methods are the most common and well-established techniques in the cybersecurity domain [36,164]. However, these algorithms may overlook undiscovered assaults or incidents due to missing features, substantial feature reduction

or poor profiling. Anomaly-based or hybrid techniques, including anomaly-based and signature-based detection techniques, can be utilized to overcome these shortcomings. To extract the target insight for a particular problem domain, such as intrusion detection, malware analysis, phishing detection and so on, a hybrid learning technique combining multiple machine learning techniques is useful. A combination of deep learning, statistical analysis and machine-learning methods can also be used to make an intelligent decision for corresponding cybersecurity solutions.

### 4.4. Feature Engineering in Cybersecurity

Due to the vast volume of network traffic data and large number of minor traffic aspects, the effectiveness and performance of a machine learning-based security model have frequently been challenged. Several techniques, such as principal component analysis, have been used to deal with the high dimensionality of data [165,166], including singular value decomposition (SVD) [167,168] and Linear Discriminant Analysis (LDA), for example. Contextual links between suspicious actions and low-level information in datasets may be useful. Such contextual data might be processed through an ontology or taxonomy for further investigation. As a result, another research challenge for machine learning approaches in cybersecurity is how to effectively choose the ideal features or extract the significant characteristics while considering machine-readable features as well as contextual features for efficient cybersecurity solutions.

### 4.5. Data Leakage

Data leakage (or leaking) happens when the training dataset contains relevant data, but similar data are unavailable or show wide variation as the models are utilized to make predictions [169]. Typically, this results in highly optimistic predictions during the model-building process, followed by the unwelcome shock of disappointing outcomes once the prediction model is put into use and tested on new data. Research in [170] identifies the problem as "leaks from the future", calling it "one of the top 10 data mining mistakes", and recommends using exploratory data analysis (EDA) to identify and eliminate leakage sources. EDA can be helpful to increase the dataset's effectiveness, thereby making the machine learning models more accurate at predicting unknown data. In recent work [171], finding and using leakage has been discussed as one of the crucial elements for winning data mining competitions, and the authors showed it to be one of the critical elements for failing a data mining application. Another study [172] describes the inclusion of giveaway qualities that forecast the target in data mining competitions since they are introduced later in the data collection process. Research in [173] provides a review of some common classifiers that are used to classify documents and datasets that have been formulated for binary prediction. To prevent leaking, researchers implemented a two-stage approach that involved marking each observation with a legitimacy tag during data collection and then observing what they referred to as a learn–predict separation. The proposed approach was significantly useful as they witnessed a maximum of 91.2% in naïve Bayes, 87.5% using k-NN and 94.2% with centroid based on various categories. In many cases, where the machine learning scientist has no control over the data collection procedure, EDA is a valuable technique for identifying leaks [174], and it could be promising for future work.

### 4.6. Homomorphic Encryption

Homomorphic Encryption (HE) is considered as one of the greatest advancements in cryptography [175]. HE provides access to a non-trustworthy third-party to process data without any clue, thereby allowing access to confidential data. The user end or the unauthorized remote server obtains access to the encrypted data and not the secret key for decryption. Hence, the host can be assured that the data are not leaked outside of the domain. HE has a long list of applications, including cloud computing, financial transactions, quantum computing threat shields, etc. HE can be applied in two ways, partial and fully. Fully Homomorphic Encryption (FHE) makes the machine learning training

process easier without data leakage. Deep learning and shallow machine learning algorithms heavily rely on domain data, which are often difficult to share publicly [176]. FHE has facilitated a new process to delegate these kinds of sensitive data sharing without sharing the actual meaningful data. One of the major drawbacks of FHE is its limitation to the use of integers [177]. Thus, researchers are trying to find matrix-based schemes for FHE. Recent research has proven effective by using the lowest degree of polynomial approximation functions such as Chebyshev with a continuous function such as sigmoid. This has created a new encryption over FHE to use in homogeneous networking [178]. Federated learning has accelerated multi-party joined learning processes by applying FHE in case of image data with sample expansions [179]. Medical and health-related data are always highly confidential since they contain Protected Health Information (PHI). One of the major breakthroughs happened when healthcare data became accessible through FHE. Researchers proved multiple effective ways to use medical images or other data from Internet of Medical Things (IoMT) using machine learning-based HE. In 2019, HE combined with chaotic mapping successfully secured data transfer, but the computational privacy was vulnerable [180]. In 2021, HE was combined with secret sharing and computation was performed on the edge computation layer. Moreover, the mathematical operation was conducted in a distributed manner, but no data leakage happened [181]. A recurrent neural network, CryptoRNN, has been introduced recently, which is mostly focused on the privacy preservation of blockchain technology [182]. Cloud-based FHE integration is the most advanced and commonly used because of the versatility of domain data access and vast computational power. The Machine Learning as a Service Platform (MLaaS) provides a wide variety of machine learning algorithms to enable FHE to protect confidential data [183]. In the early exploration of HE in Wireless Sensor Networks (WSNs), researchers experimented with the performance of HE in a network simulation NS-2 tool, where, for each experimental agent, the environment remained similar. For data aggregation operation, FHE consumed less time than DAA, which decrypts hop-by-hop, and achieved the time complexity of O(n) [184]. HE increases the global data stream and machine learning's practical applications by scaling, along with enhancing the overall cybersecurity [185].

### 4.7. Quantum Computing

While in their infancy, quantum computers were once established as having the potential to break the security offered by asymmetric encryption techniques [186]. Asymmetric key encryption relies on public and private keys. These keys are generated by factoring two extremely large prime numbers. Factoring small prime numbers is possible but keys that are very large can take thousands of years to decrypt, making our data secure. Shor's algorithm [187] provides an alternative solution to factor these large prime numbers, but, again, it is slow. Quantum computing, with its principal of superposition, can rapidly derive the factors at a fraction of the time that a binary computing system would take. This makes algorithms such as RSA and DES, elliptic curve algorithms such as ECDSA and digital signature algorithms no longer secure. The authors in [188] mentioned that to break a 56-bit DES, Grover's algorithm [189] utilizing quantum computing would only require 185 searches for key identification. Symmetric key algorithms such as AES are still resistant to quantum computing. Researchers are exploring both quantum and mathematical techniques to circumvent these limitations. An example is the BB84 protocol [190], which is a type of quantum key distribution. Mathematical approaches such as lattice-based cryptography [191] are also being explored. While quantum computing can be detrimental to asymmetric encryption, it can also speed up machine learning if used as sub-routines [192]. This can significantly reduce the prediction time if used by algorithms such as SVM, which can require a lot of time, implementing kernel transformations to derive a hyperplane. They can also be used in deep learning if configured properly. However, there are some challenges since quantum neural networks have linear dynamics [193].

## 5. Conclusions

In this study, which was prompted by the growing importance of cybersecurity and machine learning technology, we looked at how machine learning techniques are utilized to make data-driven intelligent decision making in cybersecurity systems and services successful. A discussion about how it affects security data, both in terms of gaining insight into security occurrences and evaluating the data, has also been presented. The focus was on machine learning advancements and difficulties in the cybersecurity area. Therefore, a discussion about the security dataset and services that go with it has been presented. Our contribution also looked at how machine learning techniques might affect the cybersecurity area and the security concerns that still exist. Thus far, traditional security options receive much attention, while machine learning algorithm-based security systems receive less attention. The study uses an IDS classification to present the many machine learning techniques employed in this discipline, with data sources as the major theme. This is followed by an explanation of IDSs' application to various data sources using this classification. Because IDSs are designed to identify attacks, it is critical to choose the appropriate data source based on the attack's characteristics. Logs include rich semantic information that can be utilized to detect SQL injection, U2R and R2L attacks, as well as for further analysis using machine learning techniques. Packets contain communication data that can be used to detect U2L and R2L assaults. Various key issues in security analysis to show the interest of future research ideas in the domain of machine learning with cybersecurity have been discussed. As attacks evolve, so will the machine learning techniques, making this a very dynamic field. To mitigate the damage caused by cyber-attacks, constant support is necessary from not only machine learning experts but from researchers and institutions, responsible for providing the latest datasets for training, thereby making this a collective approach.

Future work will focus on a feasibility analysis of machine learning and its deployment to monitor real-time traffic. This is extremely challenging due to the diverse nature of internet packets. Another aspect that complicates the approach is that all packets are encrypted. Hence, a focus of our future research will be Homomorphic Encryption. A brief discussion on this new technique has been presented in the review, but it requires a detailed analysis. Threats revolving around quantum computing and its impact on public key encryption will also be explored.

**Author Contributions:** Conceptualization, M.A., K.E.N., R.G. and M.M.C.; methodology, M.A. and K.E.N.; software, M.A.; validation, M.A., K.E.N. and R.G.; data curation, M.A., N.R., J.F.C. and K.E.N.; writing—original draft preparation, M.A., R.G., N.R., J.C., and M.M.C.; writing—review and editing, R.G., N.R. and M.M.C.; visualization, R.G., M.A., N.R. and M.M.C.; supervision, K.E.N.; funding acquisition, K.E.N. All authors have read and agreed to the published version of the manuscript.

**Funding:** This research was funded by the Department of Computer Science at North Dakota State University.

**Institutional Review Board Statement:** Not applicable

**Informed Consent Statement:** Not applicable

**Data Availability Statement:** Not applicable

**Conflicts of Interest:** The authors declare no conflicts of interest.

## Abbreviations

The following abbreviations are used in this manuscript:

RNN         Recurrent Neural Network
CNN         Convolutional Neural Network
LSTM        Long Short-Term Memory
Bi-LSTM     Bidirectional Long Short-Term Memory
GRU         Gated Recurrent Units
RF          Random Forest
NB          Naive Bayes
DoS         Denial of Service
DDoS        Distributed Denial of Service
SVM         Support Vector Machines
ICT         Information and Communication Technology
MITM        Man-in-the-Middle attack
IDS         Intrusion Detection System
FAR         False Acceptance Rate
RBF         Radial Basis Function

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
