# Peer review of "Cybersecurity Threats and Their Mitigation Approaches Using Machine Learning—A Review"

_jcp, doi:10.3390/jcp2030027_

Round 1

Reviewer 1 Report

The paper is a review to summarise several machine learning approaches in the application of the cybersecurity domain. Some suggestions are listed below:

1.     In section 3, a number of ML methods are summarised, but it seems that the authors only described the prediction accuracy of different approaches. More details could be included. Particularly, KDD 99 dataset has more than 40 features, any feature selection/extraction methods applied? Any data pre-processing process is applied? The details of each ML algorithm could also be included. The authors could refer this paper. 

Li, J., Qu, Y., Chao, F., Shum, H.P., Ho, E.S. and Yang, L., 2019. Machine learning algorithms for network intrusion detection. AI in Cybersecurity, pp.151-179

2.     In Section 4, the authors explained some current issues, such as the quality of the datasets. Although it is a review paper, it will be good if some solutions or suggestions are mentioned. 

Author Response

Dear reviewer, 

Thank you for the suggestions. They were very helpful. We have submitted a word document with all the changes. 

Reviewer 2 Report

My main concerns are the following:

  • There have been many survey papers about AI or ML in cybersecurity. What is the motivation for proposing another survey? How can this survey relate/compare to other surveys?
  • I suggest the authors focus only on the discussion about cybersecurity in Section 2, while ML application and discussion can be explained in Section 3. For example, the discussion in Section 2.2 currently mixes security and ML.
  • The authors have explained Section 2 comprehensively, but unfortunately, they do not discuss it further in Section 3. For example, the authors have described types of cyberattacks such as malware, ransomware, phishing, MITM, etc. But there is no taxonomy or discussion on how Machine Learning can be applied to solve those issues in Section 3. Similarly, the NIST 5 stages of attack are not described in Section 3 (no taxonomy concerning ML). 
  • A detailed taxonomy is essential for a survey paper. More comparisons and classifications can be extracted from the reference to attract readers with interesting insights.

My other minor comments:

  • Typo "dot" in lines 24, 125, and others; please recheck the manuscript.
  • Line 47-48, "As illustrated in Figure 1, the efficiency of various connected technologies is rapidly increasing." How can this sentence relate to Figure 1? I cannot find any "efficiency" in figure 1.
  • In lines 92-100, consider using numbers to describe Section. For example, instead of the "Background" Section, use Section 2.
  • In line 120, consider removing "The property of" so it can highlight the CIA better.
  • Line 138, "ransom is pain?"
  • Line 212 to 248. The paragraph is too long, though I understand the intention of the "one paragraph, one phase" writing approach.
  • Figure 2 is not mentioned in the paragraph.
  • Line 281-284, are the descriptions for HIDS and NIDS correct?
  • In Figure 3, why does "Machine Learning" have multiple lines to the right side of the figure?
  • I think it is better if Figure 3 matches the paragraph descriptions in Section 2.3.1. For example, lines 287-298 discuss "signature-based detection." However, there is no "signature-based detection" in Figure 3.
  • In table 1, check the horizontal line in the Firewall description.
  • Put references in Table 1 in each cell if possible.
  • Section 2.3.2 discuss malware, but previously malware has been described in lines 129-136.
  • Put references in Table 2 for each row if possible -> paper or link to the dataset.
  • From Table 3 and Table 4, I cannot find the ENRON Spam dataset in Table 2. Consider listing all datasets present in Table 3 and Table 4 to Table 2, and remove datasets that are not relevant from Table 2.
  • I think it is better if the algorithms in Table 3 appear in Figure 4. For example, I cannot find Random Forest in Figure 4; the same logic can be applied to Table 4.
  • Change the title of Section 4 to "Future/Possible improvements for ML-based cybersecurity." or other words with similar context; be more specific. Otherwise, simply use "Discussion" as the title.

Author Response

(The authors gave the same response as above.)

Reviewer 3 Report

The paper is clearly written and encompasses the targeted research area with corresponding references. However, some parts should be expanded:

1. Please expand your work with data leakage protection methods (in general, but of course with emphasis on the ones relevant for machine learning and vice versa).

2. Please expand your work with possible applications of homomorphic encryption. Namely, it enables machine learning algorithms to be carried out on encrypted data and this opens a lot of possibilities for new mitigation methods, e.g. in case of data leakage protection, but also in general.

3. Please expand your work with challenges that will arise with quantum computing.

4. For Table 2 please add corresponding references/links to databases.

Author Response

(The authors gave the same response as above.)

Reviewer 4 Report

This review paper looked at how machine learning techniques are utilized to make data-driven intelligent decision-making in cybersecurity. The authors focused on the machine learning techniques that have been implemented on cybersecurity data to make these systems secure. Here are my comments on it:

Strengths:

 (+) The paper is well-written.

 (+) The problem and subject is important in the scope of journal. 

 (+) The problem is well-motivated and well-defined.

 (+) The literature review is good, and the bibliography is sufficient and well given.

 (+) The proposed method for writing review is well-explained.

 (+) The figures generally are appropriate.

Weaknesses:

==== ENGLISH ==== 

The pronoun 'we' is used too many times in the paper. Generally, 'we' is appropriate to discuss future work in the conclusion but besides that it should be used sparingly.   

Some sentences are too long. Generally, it is better to write short sentences with one idea per sentence.

==== FIGURES ==== 

The text of some figure(s) (Figure 2 for example) is too small. Authors should make sure that the text can be read if printed on paper.

==== CONCLUSION ==== 

Some text must be added to discuss the future work or research opportunities. Basically, this the idea of review papers, so it needs to highlighted and discussed. 

Author Response

(The authors gave the same response as above.)

Round 2

Reviewer 1 Report

In Section 4, the authors explained some current issues, such as the quality of the datasets. It will be good if some solutions or suggestions are mentioned. 

Author Response

Dear reviewer, 

We sincerely apologize for the confusion. It turns out that the changes were made but we had forgotten to add them in the word file with our responses. Here it is: 

In Section 4, the authors explained some current issues, such as the quality of the datasets. Although it is a review paper, it will be good if some solutions or suggestions are mentioned.

Thank you. The following discussion has been added:

Some methods to solve these issues are associated with feature engineering \cite{ahsan2021enhancing} where model features are analyzed to remove correlated features. This technique reduces data dimensionality thereby reducing complexity. Handling data imbalance is imperative which can be done by utilizing hybrid models as reported in \cite{li2019machine} or generating synthetic data \cite{massaoudi2022intrusion, ahsan2018smote}. Other issues pertaining to data leakage should also be addressed.

We also added another section on data leakage in Section 4 to address this issue: 

Data leakage (or leaking) happens when the training data set contains relevant data, but similar data is unavailable or show wide variation as the models are utilized to make predictions \cite{kaufman2012leakage}. Typically, this results in highly optimistic predictions during the model-building process, followed by the unwelcome shock of disappointing outcomes once the prediction model is put into use and tested on new data. Research in \cite{nisbet2009handbook} identifies the problem as "leaks from the future," calling it "one of the top 10 data mining mistakes," and advises using exploratory data analysis (EDA) to identify and eliminate leakage sources. EDA can be helpful to increase the datasets effectiveness thereby making the machine learning models more accurate at predicting unknown data.  In recent work, \cite{rosset2010medical} finding and using leakage has been discussed as one of the crucial elements for winning data mining competitions and showed eventually one of the critical elements for failing a data mining application. Another study \cite{kohavi2000kdd} describes the inclusion of giveaway qualities that forecast the target in data mining competitions since they are introduced later in the data collection process. Research in \cite{gupta2022tidf} provides a review of some common classifiers that are used to classify documents and data sets that have been formulated for binary prediction. To prevent leaking, researchers implemented a two-stage approach that involved marking each observation with legitimacy tags during data collection and then observing what they referred to as a learn-predict separation. The proposed approach was significantly useful as they witnessed a maximum of 91.2\% in Naïve Bayes, 87.5\% using k-NN, and 94.2% with Centroid Based on various categories. In many cases, where the machine learning scientist has no control over the data collection procedure, EDA is a valuable technique for identifying leaks \cite{stuart1984understanding}, and it can be potential for future work.

Hope this helps. 

Reviewer 2 Report

The authors have addressed all of my previous comments well.

Please recheck the grammar/spelling and organization of the paper one more time. Make sure it is easy to follow for readers.

Author Response

Thank you for giving us the opportunity to review the paper. We went through the document once again to ensure there were no grammar or spelling issues. 

Reviewer 3 Report

The authors have successfully resolved all issues listed in the previous review.

Author Response

Thank you for giving us the opportunity to make changes to our paper.